# The IGF–PAPP-A–Stanniocalcin Axis in Serum and Ascites Associates with Prognosis in Patients with Ovarian Cancer

**DOI:** 10.3390/ijms25042014

**Published:** 2024-02-07

**Authors:** Rikke Hjortebjerg, Claus Høgdall, Kristian Horsman Hansen, Estrid Høgdall, Jan Frystyk

**Affiliations:** 1Steno Diabetes Center Odense, Odense University Hospital, 5000 Odense, Denmark; 2Department of Clinical Research, University of Southern Denmark, 5230 Odense, Denmark; jan.frystyk@rsyd.dk; 3Department of Gynecology, Juliane Marie Center, Rigshospitalet, 2100 Copenhagen, Denmark; claus.hogdall@regionh.dk; 4Endocrine Research Unit, Department of Endocrinology, Odense University Hospital, 5000 Odense, Denmark; khorsman@health.sdu.dk; 5OPEN Lab, Odense University Hospital, 5000 Odense, Denmark; 6Department of Pathology, Herlev University Hospital, 2730 Herlev, Denmark; estrid.hoegdall@regionh.dk

**Keywords:** ascites, insulin-like growth factor, insulin-like growth factor binding protein, mortality, ovarian cancer, pregnancy-associated plasma protein-A, prognosis, stanniocalcin

## Abstract

Pregnancy-associated plasma protein-A (PAPP-A) and PAPP-A2 modulate insulin-like growth factor (IGF) action and are inhibited by the stanniocalcins (STC1 and STC2). We previously demonstrated increased PAPP-A and IGF activity in ascites from women with ovarian carcinomas. In this prospective, longitudinal study of 107 women with ovarian cancer and ascites accumulation, we determined corresponding serum and ascites levels of IGF-1, IGF-2, PAPP-A, PAPP-A2, STC1, and STC2 and assessed their relationship with mortality. As compared to serum, we found highly increased ascites levels of PAPP-A (51-fold) and PAPP-A2 (4-fold). Elevated levels were also observed for IGF-1 (12%), STC1 (90%) and STC2 (68%). In contrast, IGF-2 was reduced by 29% in ascites. Patients were followed for a median of 38.4 months (range: 45 days to 8.9 years), during which 73 patients (68.2%) died. Overall survival was longer for patients with high serum IGF-1 (hazard ratio (HR) per doubling in protein concentration: 0.60, 95% CI: 0.40–0.90). However, patients with high ascites levels of IGF-1 showed a poorer prognosis (HR: 2.00 (1.26–3.27)). High serum and ascites IGF-2 levels were associated with increased risk of mortality (HR: 2.01 (1.22–3.30) and HR: 1.78 (1.24–2.54), respectively). Similarly, serum PAPP-A2 was associated with mortality (HR: 1.26 (1.08–1.48)). Our findings demonstrate the presence and activity of the IGF system in the local tumor ecosystem, which is likely a characteristic feature of malignant disease and plays a role in its peritoneal dissemination. The potential clinical implications are supported by our finding that serum levels of the proteins are associated with patient prognosis.

## 1. Introduction

Ovarian cancer, the most lethal gynecological malignancy, preferentially metastasizes into the peritoneum, resulting in often limited and unspecific symptoms and advanced disease at diagnosis [1]. The accumulation of malignant ascites in the abdominal cavity often accompanies and plays an important role in intraperitoneal dissemination, and its presence signifies a dismal prognosis [2]. A variety of soluble factors present in ascites contribute to a unique proinflammatory and carcinogenic microenvironment that fuels tumor growth, survival, and metastatic voyage [1]. Among these are members of the insulin-like growth factor (IGF) system, which are reported as overexpressed in both tissue and ascites from patients diagnosed with ovarian cancer [3,4,5]. IGF-1 stimulates most steps of cancer progression by signaling through the IGF-1 receptor (IGF-1R) [6,7,8], and IGF-2 promotes tumor growth through binding primarily to the insulin receptor subtype A (IR-A). High gene and protein expressions of the IGF-1R and IR-A are described in several cancers, including ovarian cancer [9,10].

The biological activity of the IGFs is modulated by a family of IGF-binding proteins (IGFBPs) as well as IGFBP proteases. Pregnancy-associated plasma protein-A (PAPP-A) and PAPP-A2 constitute the pappalysin family of metalloproteases [11]. They have been shown to play key roles in regulating IGF signaling in cancers [12,13,14,15,16]. The PAPP-As are responsible for the proteolytic cleavage of specific IGFBPs, through which they increase tissue concentrations of free IGF, which becomes available for receptor activation [17,18,19]. In breast cancer, an elevated serum PAPP-A has been shown to be associated with low recurrence-free and overall survival [20], and recently, we demonstrated that serum PAPP-A2 is higher in patients with lung cancer and predictive of all-cause mortality [13]. In 2015, stanniocalcin-1 (STC1) and -2 (STC2) were discovered as potent inhibitors of PAPP-A and PAPP-A2 [21,22]. The STCs have oncogenic properties and levels are altered in multiple cancer types [23,24,25]. Thus, the regulation exerted by the PAPP-As and STCs on the IGF signaling cascade is indubitably an accomplice in cancer progression.

We previously investigated PAPP-A levels in serum and ascites from 22 women diagnosed with ovarian cancer [4]. In corresponding samples, PAPP-A was significantly increased in ascites as compared to serum from the same patient, and we observed an increased PAPP-A-mediated proteolysis of IGFBP-4, which is the principal substrate for PAPP-A [26,27]. In addition, it was shown that the ability of ascites to activate the IGF-1R in vitro was increased by 31% as compared to serum [28]. Thus, the study supported the concept that PAPP-A may increase tumor growth via its ability to promote IGF-1 action in the local tumor milieu.

Interestingly, proteins highly enriched in ascites may represent a potential source of circulating markers for ovarian cancer by translocation from the ascites to the blood circulation. Thus, our findings gave rise to the current study, which evaluated the potential clinical significance of the IGF system by associating ascites and circulating levels with all-cause mortality in 107 patients with ovarian cancer. Furthermore, we aimed to extend our previous investigation and compare levels of IGF-1, IGF-2, PAPP-A, PAPP-A2, STC1, and STC2 in corresponding ascites and serum samples, providing new insights into the mechanisms behind IGF-mediated tumorigenicity and cancer biology.

## 2. Results

### 2.1. Study Cohort and Characteristics

Of the 107 ovarian cancer patients, 49 (45.8%) had primary surgery, 41 (38.3%) had interval surgery after 3 series of chemotherapy, and 17 (15.9%) were only treated with chemotherapy. Chemotherapy followed Danish guidelines consisting of Paclitaxel and Platin-based regimes. The neoadjuvant and oncologic treated only patients were generally older and had entered menopause, and they presented with more advanced-stage cancer. In operated patients, complete macro-radical tumor removal was intended at surgery. It was successfully achieved in 70.5%, whereas 29.5% had residual tumors.

According to histology, the cases of ovarian cancer included 94 serous adenocarcinomas, four endometrioid adenocarcinomas, four mucinous adenocarcinomas, one clear cell carcinoma, and four carcinosarcomas. In addition, 7 patients presented with borderline tumors and 14 presented with other cancer types. In addition to baseline characteristics, these patients were not included in subsequent analyses. The accumulation of ascites was a requirement for patient inclusion, and this occurs most frequently in patients with advanced-stage ovarian cancer. Therefore, 89.7% of ovarian cancer patients presented with FIGO stage III or IV ovarian cancer at diagnosis, meaning that the ovarian cancer had spread outside the pelvis into the abdominal cavity, to lymph nodes, or other body organs. Neither histology nor stage was associated with age or body mass index (BMI) at diagnosis. The clinical characteristics of the patients are shown in Table 1.

### 2.2. IGF Levels in the Circulation and Ascites from Patients with Ovarian Cancer

In ascites, the total IGF-1 level was increased by 12% as compared to the levels of IGF-1 measured in serum, whereas IGF-2 was reduced by 29% in ascites (Table 2 and Figure 1). PAPP-A and PAPP-A2 were highly abundant in ascites with levels being augmented by 51-fold and 4-fold, respectively. The PAPP-A and PAPP-A2 inhibitors STC1 and STC2 were increased by 90% and 68%, respectively. To corroborate previous findings [4], we used a subset of 20 patients from the total cohort to perform measurements of IGF bioactivity, IGFBP-4, and IGFBP-4 fragments. As expected, we found a reduction of 77% in intact IGFBP-4 and an increase in the PAPP-A-generated IGFBP-4 fragments of more than 60%. IGF bioactivity was 172% higher in ascites than in serum.

Since most patients (87.9%) had ovarian cancer of the serous adenocarcinoma subtype, the remaining subtype groups were of insufficient size to allow for histology-specific analyses of the IGF system proteins. Similarly, as most patients presented with stage III or IV cancer, separate analyses were not performed according to stage.

### 2.3. Relations between IGF Protein Levels

In unadjusted correlation analyses, significant associations between ascites and serum concentrations of the same protein were observed for all biomarkers except IGFBP-4 and PAPP-A (Table 3).

Total IGF-1 was positively associated with IGF bioactivity in both serum and ascites. Interestingly, its association with the PAPP-A-generated IGFBP-4 fragments was negative in the circulation but positive in ascites. In ascites, high IGFBP-4 fragmentation was also related to high IGF bioactivity. STC2 was significantly associated with levels of IGF-1 in serum and with PAPP-A in both serum and ascites. In addition, high STC1 and STC2 were correlated with low levels of bioactive IGF and high levels of intact IGFBP-4 in ascites. Overall, statistically significant correlations were more frequently observed in ascites than in the blood fractions.

To investigate the association between total IGF-1 and IGF bioactivity in serum versus ascites, we calculated the percentage of bioactive IGF ((IGF bioactivity/total IGF-1) × 100) in each compartment. In ascites, the fraction of IGF-1 that appeared to be bioactive was 93% higher than in serum (2.9 (2.1–4.3)% versus 1.3 (1.1–1.6)%, *p* < 0.001).

Serum CA125 level and patient RMI were positively associated with serum PAPP-A2 (r = 0.215, *p* < 0.05, and r = 270, *p* < 0.005, respectively) but did not correlate with other proteins.

### 2.4. Survival Analyses of Ovarian Cancer Patients

The median (range) follow-up of all cancer patients was 38.4 months (45 days to 8.9 years), during which 73 patients (68.2%) died (Table 1). Overall mortality differed according to subtype at diagnosis; 29% for patients with stage I ovarian cancer, 50% for stage II, 69% for stage III, and 78% for stage IV (*p* < 0.05). Median survival was 50 months for patients with stage I ovarian cancer, 42 months for stage II, 40 months for stage III, and 26 months for stage IV (*p* < 0.05).

The associations between all-cause mortality and the various proteins dichotomized at the median value (<median or ≥median) were investigated using log-rank analysis (Table 4). Patients with serum IGF-1 levels below the median had reduced overall survival as compared to patients with serum IGF-1 above the median. However, analysis of IGF-1 in ascites showed the opposite association; mortality was increased in patients with high IGF-1 levels. In both the circulation and ascites, IGF-2 level above the median was associated with reduced survival. High PAPP-A2 serum level was associated with increased mortality, whereas no association was found in ascites. PAPP-A, STC1, and STC2 were not associated with outcome.

Corresponding Kaplan–Meier survival curves were generated for proteins with significant associations in the log-rank analysis of either the circulation or ascites (Figure 2).

The ability of the proteins as continuous predictors for mortality was further assessed. ROC AUC values for serum and ascites IGF-1 levels were 0.63 (0.51–0.74) and 0.66 (0.55–0.78), respectively. ROC AUC for serum IGF-2 was 0.67 (0.56–0.77), and that of ascites IGF-2 was 0.65 (0.54–0.77). Serum PAPP-A2 showed an ROC AUC of 0.64 (0.53–0.76). In comparison, ROC models using serum CA125 and patient RMI did not provide any discriminatory capacity (AUC 0.52 (0.40–0.65) and 0.50 (0.38–0.63), respectively).

In Cox regression analyses, associations with mortality during follow-up were investigated for all proteins using continuous log_2_-transformed variables (Figure 3). Levels of IGF-1 and IGF-2 in serum and ascites were all associated with mortality in univariable analyses. Following multivariable adjustments (model 1), both proteins remained associated with the outcome. When further adjusting for CCI and performance score (model 2), the association between low serum IGF-1 level and mortality was no longer significant. The serum level of PAPP-A2 was associated with mortality in both univariable and multivariable analyses. No associations were observed for PAPP-A or STC1. The STC2 level in ascites appeared associated with mortality but only in the multivariable analyses. All analyses were stratified to the histological subgroup of 94 patients with serous adenocarcinomas, and similar results were obtained.

## 3. Discussion

In this prospective longitudinal study of women with ovarian tumors, we demonstrate a local presence and activity of the IGF system in ascites that differs substantially from that of the circulating IGF system, and furthermore, show the potential of IGF-1, IGF-2, and PAPP-A2 in predicting patient outcomes. This is the first study to suggest a role of PAPP-A2 in ovarian cancer and to demonstrate a prognostic value of IGF-1 and -2 in ascites. The results encourage investigations into the potential of these proteins as markers of disease or of treatment effects, e.g., that of Bevacizumab, which is a monoclonal antibody routinely used in the treatment of advanced ovarian cancer.

The present study builds upon previous original findings in 22 patients [4], where we demonstrated an increased PAPP-A and IGF activity in the local tumor milieu. Our data led us to hypothesize that the high IGF activity in ascites could be a characteristic feature of malignant disease, play a significant role in its peritoneal dissemination, and potentially be associated with a worse patient prognosis. Thus, in the present study, we evaluated a parallel analysis of the IGF proteins in blood and ascites and investigated their prognostic values.

The buildup of ascites within the abdomen is a hallmark of ovarian cancer and occurs as malignant cells secrete hormones and cytokines that cause vascularization, increased fluid filtration, and lymphatic blockade [1,2]. Thus, malignant ascites reflects a microenvironment of shed and secreted proteins by ovarian cancer, and it composes an ecosystem that acts as the main route for metastases. Accordingly, ascites constitutes an ideal reservoir for the identification of measurable factors indicative of ovarian cancer risk, occurrence, recurrence, or patient outcome [29,30]. In this study, all IGF proteins were measurable in the extravascular ascites fluid, verifying that all necessary components for an IGF-mediated autocrine/paracrine loop were present in the cancer microenvironment. Except for IGF-2 and IGFBP-4, the concentration and activity of all proteins and enzymes exceeded those of serum, indicating that the proteins to some extent reflect the malignant secretome. Especially for PAPP-A and PAPP-A2, being increased by 51-fold and 4-fold in ascites as compared to serum, respectively, it is unlikely that the proteases originate from the intravascular space and traverse into the abdominal cavity; more likely, they are of ovarian cancer origin. The high amount of PAPP-A is also expected to be responsible for the increased IGFBP-4 degradation and concomitantly increased IGF bioactivity in ascites. Despite total IGF-1 levels being relatively comparable in serum and ascites, the IGF bioactivity was increased by 172% in ascites, and the relative proportion of bioactive IGF compared to total IGF-1 was almost doubled in ascites as compared to the circulation. These results agree with our previous report, in which we demonstrated that in ascites, PAPP-A was responsible for the degradation of IGFBP-4 and generation of IGFBP-4 fragments [4]. It is noteworthy that we also demonstrated that the PAPP-A mediated the liberation of free IGF augmented IGF-1R activation and phosphorylation of downstream intracellular IGF-1R signaling proteins [4].

The PAPP-A2 level in the circulation possessed prognostic power and was associated with shorter survival. This novel finding supports that PAPP-A2 is likely to influence the IGF signaling cascade in ovarian cancers. Interestingly, despite the high level of PAPP-A in ascites, this enzyme was not associated with increased mortality. This result agrees with a recent study in patients with lung cancer, which demonstrated the presence of PAPP-A2 in cancer tissue, showed elevated circulating PAPP-A2 levels, and established an association between PAPP-A2 and all-cause mortality [13]. Such associations were not observed for PAPP-A. Interestingly, PAPP-A has repeatedly been described in the context of cancer, whereas studies on PAPP-A2 in human pathologic conditions outside pregnancy are still limited. For example, it has been shown that murine PAPP-A deficiency results in a delayed occurrence of tumors [31,32]; that PAPP-A level is elevated in lung cancer patients [33] and a prognostic marker in breast cancer [20]; and that PAPP-A is expressed by a wide range of cells of malignant origin [4,15,34]. On the contrary, PAPP-A2 was only recently demonstrated to possess relevant functions in human physiology. A novel loss-of-function mutation in the human *PAPPA2* gene was shown to give rise to children with short stature and severe perturbations in the IGF system [35]. This was the first description of a human condition with reduced IGF-1 activity caused by defects in IGFBP regulation. However, PAPP-A2 has only recently emerged as a suspected accomplice in neoplasia, and reports on PAPP-A2 in ovarian cancer are non-existent. Therefore, this novel connection between PAPP-A2 and ovarian cancer calls for functional studies to investigate a potential causal relationship as well as further validation in other cohorts.

Serum PAPP-A levels did not correlate with the corresponding ascites levels, suggesting limited spill-over from ascites. The movement of proteins that are highly enriched in the peritoneal cavity into the intravascular space depends on the permeability of the capillary endothelium and the peritoneal membrane [36], and efflux/influx across the pores correlate with molecular weight; smaller proteins can easily diffuse into the capillary bed, whereas the transport of larger molecules is generally prevented [37]. PAPP-A is primarily bound to other proteins and circulates as a large complex of 500 kDa, whereas the vast majority of PAPP-A2 is not covalently bound to other molecules and therefore is of smaller weight [38,39]. Thus, the high-molecular weight of PAPP-A may hinder paracellular relocation to the circulation and limit its potential as a circulating biomarker.

Ovarian cancer patients with high ascites levels of IGF-1 or IGF-2 showed a poorer prognosis reflected in shorter overall survival. The association between high IGF-2 and the risk of death was also observed in serum, in which the mortality HR increased by >75% with each 2-fold increase in IGF-2. By signaling through the IGF-1R and the IR-A, the IGFs fuel malignant disease and play a role in its dissemination [6,7,8], so it comes as no surprise that an augmented ascites IGF level is associated with a worse clinical outcome in patients. Supportive of this, it has previously been demonstrated that the IGF-1R and IR-A are highly expressed in ovarian cancer cell lines and tissues [40,41,42]. Furthermore, in ovarian cancers, insulin receptors are preferentially overexpressed as IR-A isoforms, which specifically favors binding by IGF-2 [41,43,44]. In contrast to the apparent malignant effects of IGF-1 in the tumor microenvironment, high levels of IGF-1 in the circulation were associated with more favorable outcomes. With every doubling in IGF-1 serum level, the mortality HR was reduced by 40%. This inverse correlation agrees with some previous findings [45,46,47] but differs from other reports [48,49]. An extensive outcome-wide analysis of cancer risk from the UK Biobank has established an inverse association between IGF-1 and the risk of ovarian cancer [50]. The discrepancy between circulating IGF-1 levels and those in the tumor microenvironment may be explained by the distinct autocrine/paracrine and endocrine influences of IGF-1. More than 80% of the circulating IGF-1 pool originates from the liver [51], and thus, any secretory contribution from an ovarian cancer may be unlikely to influence endocrine IGF-1. Instead, serum levels reflect numerous conditions related to overall human health status. The secretion of IGF-1 declines continuously with age, and low IGF-1 is associated with impaired nutritional status and weight loss, frailty, sarcopenia, osteoporosis, and inflammation [52,53]. Many of these states are coexistent with diseases, including cancer, and suggest that the inverse association may be the result of reverse causality. Potential metastasis to the liver may also impair IGF-1 production. This may explain why after extensive adjustments for confounding variables in the multivariable regression (model 2), IGF-I was no longer associated with outcome.

We demonstrated an excess abundance of STC1 and STC2 in ascites as compared to the circulation, and STC2 correlated with the PAPP-A level in both compartments. Similar associations have previously been demonstrated in serum and pericardial fluid from patients with cardiovascular disease [27].

However, despite elevated STC1 and STC2 ascites levels, they were unable to block the increased PAPP-A activity, as judged by the high IGFBP-4 fragmentation. Healthy ovaries express both STC1 and STC2, and levels are increased in ovarian cancer compared to non-cancerous tissue [24,25]. This supports our hypothesis that the high STC concentration in ascites derives from the tumor. Therefore, it is tempting to speculate that the PAPP-As and STCs may be co-expressed in ovarian cancers. Multiple reports also link the STCs to other human cancers, although both up- and down-regulations have been observed [23]. Among others, they have been shown to associate with cellular immortalization, be overexpressed in tumors, stimulate cancer migration and invasion, and reflect disease severity and prognosis [24,25]. However, in this study, neither STC1 nor STC2 were able to prognosticate mortality, and we found no evidence to support the ability of the STCs to act as biomarkers of outcome in patients with ovarian cancer. Of note, the STCs are multifunctional proteins with the ability to interact with other regulatory pathways apart from the IGF system [18], and their primary involvement in cancer may be through IGF-independent mechanisms. Thus, the mechanisms by which the STCs participate in pathophysiological processes in cancer remain enigmatic.

Today, the most widely used serum marker for ovarian cancer is CA125, but its utility as a screening marker is limited because of high false positive rates and elevation in other malignant and non-malignant conditions [54]. Nevertheless, CA125 is used for treatment monitoring and surveillance for ovarian cancer recurrence. In this cohort, increased serum CA125 as well as patient RMI at diagnosis was not related to mortality in the ROC analysis or in univariable or multivariable survival analyses.

Unfortunately, the size of the current discovery cohort does not allow for a more in-depth investigation into the biomarker potential of the IGF proteins. Thus, our results cannot necessarily be extrapolated to other patient cohorts. Clinical adoption will await further investigation of their performances in larger, independent validation cohorts to verify their potential clinical utilities, sensitivities, and specificities. Finally, it must always be determined to what extent a new biomarker provides any prognostic or predictive value beyond that of conventional clinical risk factors.

Certain limitations in the study should be acknowledged. The production of ascites was a prerequisite for patient inclusion, and as this occurs more frequently in women with advanced-stage cancer, our patient group may not be representative of a general ovarian cancer cohort. Of our 107 patients, 89.7% presented with stage III or IV disease at diagnosis, whereas this is 75% in ovarian cancer patients in general [55]. Accordingly, any protein that may hold potential as a marker of early-stage disease would not be identified in this study cohort. Finally, it would have been ideal to include an age-matched cohort of healthy women to compare circulating protein levels. However, the levels of most of the proteins were determined in controls in our original study of the 22 ovarian cancer patients, and results can be found in there [4].

## 4. Materials and Methods

### 4.1. Study Cohort

A total of 128 patients were enrolled in the study (Figure 4). All patients were recruited from the prospective, ongoing Pelvic Mass Study, initiated in 2004 at the Gynecologic Department, Rigshospitalet, Copenhagen, Denmark. Biological materials were collected through the Danish CancerBiobank, which aims to collect high-quality blood, tissues, and clinical and epidemiological data on all patients admitted for surgery because of a tumor in the female pelvis. Patients were examined according to the Danish Cancer Fast Track Guidelines and admitted to surgery or biopsy for diagnosis. All patient tissues were examined by specialized pathologists. In the current study, patients were included if they had a primary tumor of ovarian origin and a volume of ascites considered pathological (>5 mL). Patients were excluded if they were unable to give informed consent, had no ascites accumulation, were pregnant, or had a known relapse of previous cancer or other active cancer. Post-inclusion, 21 patients were revealed to have other types of tumors. These diagnoses included cervix cancer (*n* = 1), endometrial cancer (*n* = 5), teratoma (*n* = 1), neuroendocrine carcinoma (*n* = 2), cancer of unknown origin (*n* = 5), and borderline tumor (*n* = 7). The patients with borderline tumors or other types of cancers were analyzed separately but otherwise excluded from the study. Thus, the final ovarian cancer cohort consisted of 107 patients.

The patients were registered in the nationwide Danish Gynecologic Cancer Database (DGCD), which is linked to the Danish Register of Causes of Death and the Danish National Patient Registry, from where regularly updated information on clinical data including serum levels of CA125, treatment, and survival of the patients can be retrieved. The Danish Register of Causes of Death collects information on all deaths in Denmark, originating from mandatory death certificates, including the date and cause of death coded according to the International Classification of Diseases, Tenth Revision (ICD-10). Using ICD-10 disease codes, information regarding the Charlson Comorbidity Index (CCI) conditions at baseline was also obtained. Because of the high-quality Danish registration system, no patients were lost during follow-up.

Blood and ascites samples were collected either at or before surgery or by paracentesis before the start of neoadjuvant chemotherapy. Samples were collected a maximum of 14 days before the start of treatments. Serum and EDTA-plasma samples were centrifuged at 4 ºC for 10 min at 2000× *g*. The supernatant was aliquoted and stored at −80° until further analyses.

Written informed consent was obtained from all patients, and the study was approved by the Danish National Committee for Research Ethics, Capital Region (approval codes; KF01-277/03 and KF01-143/04, H15020061) and conducted per the guidelines of the Declaration of Helsinki

### 4.2. Patient Follow-Up

The first patient was included on 12 June 2013, and the last patient was included on 24 April 2019. Patients were followed from the date of referral to the Gynecologic Department until death of any cause or 22 August 2022. Thus, all patients were followed for at least 40 months.

### 4.3. Laboratory Measurements

Routine biochemistry was performed at the hospital’s laboratory using widely available automated assays.

The Risk of Malignancy Index (RMI) score was calculated as the product of the ultrasound scan score, the menopausal status (premenopausal coded 1, postmenopausal coded 3), and the serum cancer antigen 125 (CA125) level. A RMI score above 200 is considered as high risk of malignancy in Denmark.

IGF-1 serum levels were measured using an IDS-iSYS Multi-Discipline Automated Analyzer (Immunodiagnostic Systems Nordic SA, København, Denmark). IGF-2 (Cat# AL-131), PAPP-A (Cat# AL-101), PAPP-A2 (Cat# AL-109), and STC2 (Cat# AL-143) levels were determined in EDTA plasma by commercial ELISA kits from AnshLabs (Webster, TX, USA). Assay procedures were as described by the manufacturer, and all assays behaved linearly within the analytical range. The STC1 EDTA-plasma level was measured using a Duoset (Cat# DY2958) from R&D Systems (Abingdon, UK). In all assays, serum or EDTA-plasma samples were analyzed in pairs with ascites in a blinded fashion with patients in random order.

To confirm previous findings, serum levels of bioactive IGF and EDTA-plasma levels of IGFBP-4 and IGFBP-4 fragments were determined in the first 20 patients from the cohort [4]. The ability of IGF to activate the IGF-1R in vitro (bioactive IGF) was determined by an in-house kinase receptor activation assay (KIRA) as originally described [28] with slight modifications, using the commercial phospho-IGF-1R ELISA kit from R&D Systems (Cat# DYC 1770E). The assay measures the capability of IGF to phosphorylate the IGF-1R in an in vitro-based model using IGF-1R gene-transfected human embryonic kidney cells. A serial dilution of rhIGF-1 (WHO 02/254) served as the calibrator. The assay signal primarily reflects the binding of IGF-1 to the IGF-1R but also the binding of IGF-2 and pro-IGF-2 (cross-reactivity 12% and 2%, respectively). Hence, the assay signal is referred to as IGF bioactivity. The KIRA assay detection limit was <0.08 µg/L, and in this study, intra- and inter-assay CVs were <12% and <20%, respectively.

EDTA-plasma levels of intact IGFBP-4 and the two PAPP-A generated fragments, C-terminal (CT)-IGFBP-4 and N-terminal (NT)-IGFBP-4, were measured by in-house time-resolved immunofluorometric assays using monoclonal antibodies and recombinant human calibrators from HyTest Ltd. (Turku, Finland). The assays were essentially performed as previously described [27,56]. In each fragment assay, one of the antibodies specifically recognizes the proteolytic neoepitope generated by PAPP-A. Detection limits were 0.5 µg/L for IGFBP-4, 0.4 µg/L for CT-IGFBP-4 and 0.9 µg/L for NT-IGFBP-4. Intra- and inter-assay CVs were <10% and <15%, respectively.

### 4.4. Statistics

The assumption of normality was checked using quantile–quantile plots, and non-normally distributed variables were log_2_-transformed prior to statistical analyses. Groups were compared using Student’s *t*-test (two groups) or one-way ANOVA and post hoc tests with Bonferroni’s correction (multiple groups). If there was evidence against the assumption of equal variance by Bartlett’s test, or if data did not follow a normal distribution, a Wilcoxon rank-sum test or Kruskal–Wallis test was applied, respectively. Categorical variables were evaluated by Fisher’s exact test or χ2-test. A test for linear trends across ordered groups was performed by linear regression analysis with the ordered group as a continuous explanatory variable with equal distance between steps.

The area under the receiver operating characteristic (ROC) curve (AUC) was used to assess the prognostic ability of the proteins as continuous variables. Kaplan–Meier survival curves were generated based on the median split of the proteins, and incidence distributions were compared using the log-rank test. Cox proportional hazard models were developed to explore associations between mortality and the continuous explanatory variables. The continuous variables were log_2_-transformed, and accordingly, one unit increase in protein level on the log_2_-scale corresponded to a doubling in protein. Hazard ratios (HRs) estimated the risk of death in an unadjusted model and a multivariable model 1 after adjustment for covariables defined a priori: age, International Federation of Gynecology and Obstetrics (FIGO) tumor stage, and macro-radical tumor removal. The age variable was dichotomized using age at diagnosis of <70 years or ≥70 years as the cut-off, as recent studies of ovarian cancer have agreed on a limit between younger and older patients of 70 years [57]. Tumor stage was also dichotomized (stage I/II versus stage III/IV). The macro-radical tumor removal variable was coded as 1 (patients with complete macro-radical tumor removal), 2 (patients with residual tumor following surgery), and 3 (patients with advanced disease that did not undergo surgery). Finally, a multivariable model 2 further included CCI and performance status as covariables. CCI was dichotomized (CCI = 0 versus CCI > 0). Performance status, a standardized estimate of a patient’s ability to perform activities of daily living, ranged from 0 to 3 (dichotomized using 0/1 versus 2/3) [58]. Of note, our proteins of interest were not independently associated with CCI and performance status, and thus, the estimated HRs in model 2 would possibly be biased toward the null (that is, would potentially represent an over-adjustment), thus producing an underestimate of the association between the marker and the outcome.

The validity of the proportional hazards and linearity assumptions were checked by log–log plots, fitted survival curves, and smoothed martingale and Schoenfeld residual plots; no deviations from proportionality were identified [59,60].

Results are presented as median (25th percentile; 75th percentile) or median (lower range value to upper range value). Categorical variables are indicated as number (n) and percentage (%) of patients. ROC AUC and HRs are presented with 95% confidence intervals (CI). Two-tailed *p* values < 0.05 were considered statistically significant. Data were analyzed using Stata software version 18 (StataCorp LP, College Station, TX, USA).

## 5. Conclusions

In patients with ovarian cancer, ascites contains an increased concentration and activity of the IGF system, which is likely a characteristic feature of malignant disease. The potential clinical implications are supported by the finding that serum levels of IGF-1, IGF-2, and PAPP-A2 are associated with patient prognosis and may find use as biomarkers for ovarian cancer.

## Figures and Tables

**Figure 1 ijms-25-02014-f001:**
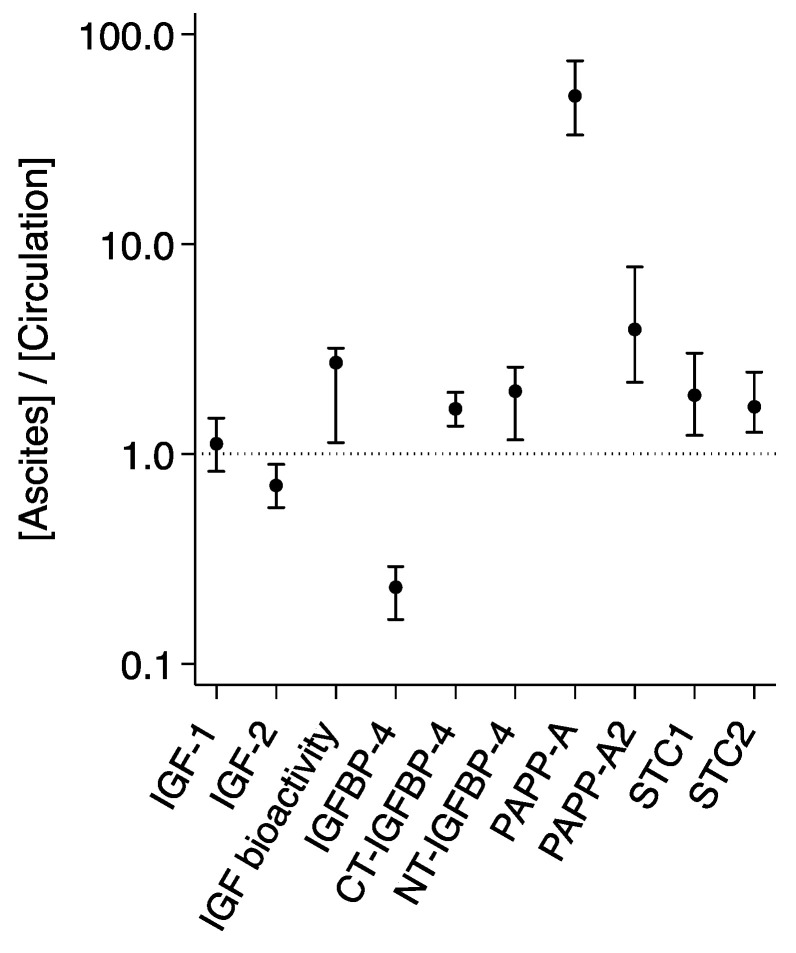
Distribution of IGF proteins in ascites and the circulation. Ratios ([Ascites]/[Circulation]) are based on measurements in 107 patients with ovarian cancer. For IGF bioactivity, intact IGFBP-4, CT- and NT-IGFBP-4, ratios are based on measurements in 20 patients. Ratio > 1 reflects a higher ascites level, and ratio < 1 reflects a higher level in the circulation. The horizontal line shows the level at which ascites = plasma. Ratios are median (25th percentile; 75th percentile) on a log scale. Differences between compartments were significant for all proteins. CT, C-terminal; IGF, insulin-like growth factor; IGFBP, IGF binding protein; NT, N-terminal; PAPP-A, pregnancy-associated plasma protein-A; STC, stanniocalcin.

**Figure 2 ijms-25-02014-f002:**
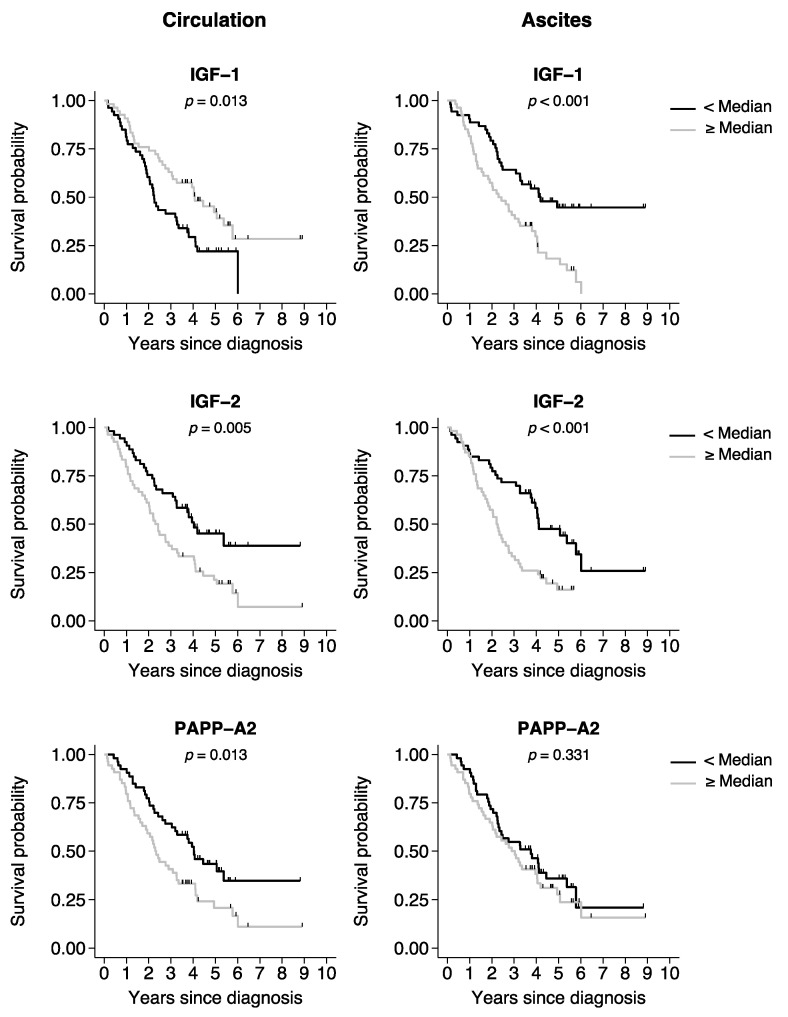
Kaplan–Meier curves of overall survival in patients with ovarian cancer. Kaplan–Meier survival curves were generated based on the median split of the protein levels in the circulation or ascites. Black line curves represent the group of patients with a protein level below the median. Gray line curves represent the patients with a protein level above the median. *p*-values: log-rank test for survival equality in the two groups. Tick marks represent censored events. IGF, insulin-like growth factor; PAPP-A2, pregnancy-associated plasma protein-A2.

**Figure 3 ijms-25-02014-f003:**
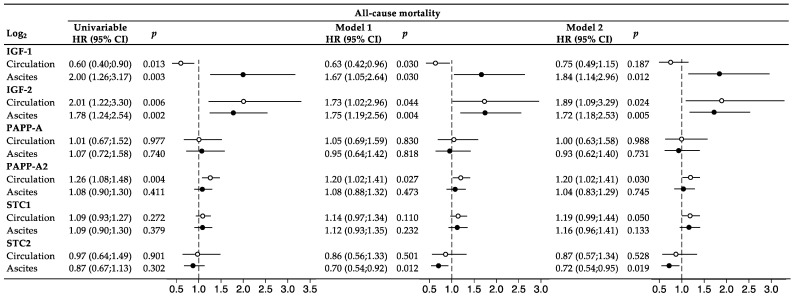
Cox regression analysis and forest plot. The IGF system markers were investigated both in univariable analyses and in multivariable analyses. Model 1: adjusted for age, tumor stage, and macro-radical tumor removal. Model 2: adjusted for covariables in model 1 + CCI and performance score. The IGF variables were log_2_-transformed, and accordingly, one unit increase on the log_2_-scale corresponded to a doubling in protein. Forest plots display HR (95% CI), with hollow circles referring to circulating levels and solid circles referring to ascites levels. The vertical dashed lines represent the points of no effect (HR = 1.00). CI, confidence interval; HR, hazard ratio; IGF, insulin-like growth factor; PAPP-A, pregnancy-associated; STC, stanniocalcin.

**Figure 4 ijms-25-02014-f004:**
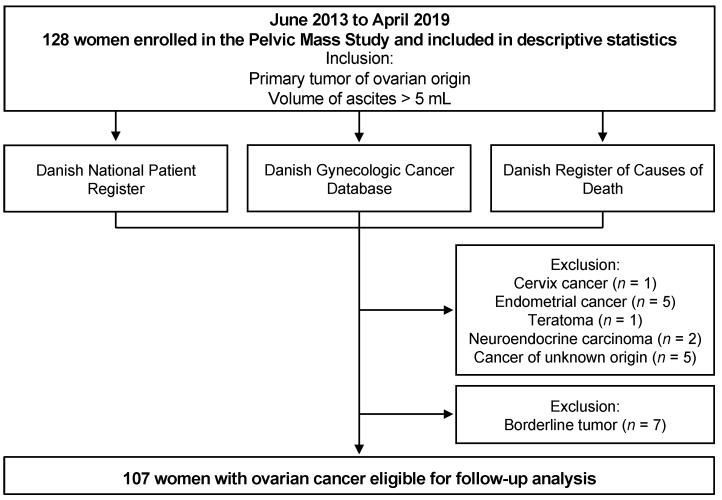
Flow chart of data collection in the study.

**Table 1 ijms-25-02014-t001:** Patient characteristics.

Characteristics	Ovarian Cancer	Other Cancer	Borderline Ovarian Tumor	*p*
Number, *n*	107	14	7	
Age, years	67.2 (56.8–74.0)	62.6(55.0–70.1)	55.2 (43.4–58.1) *	0.019
BMI, kg/m^2^	23.8 (21.1–27.5)	25.5 (22.4–29.8)	35.9 (26.6–43.9) *	0.017
Treatment, *n* (%)				
Primary surgery	49 (58.8)			
Interval surgery	41 (38.3)			
Chemotherapy	17 (15.9)			
Postmenopausal, n (%)				
Yes	92 (86.0)	10 (71.4)	4 (57.1)	0.050
No	15 (14.0)	4 (28.6)	3 (42.9)	
Smoker, n (%)				
Never	49 (45.8)	6 (50.0)	4 (57.1)	0.408
Former	41 (38.3)	5 (41.7)	2 (28.6)	
Current	16 (15.0)	1 (8.3)	0 (0.0)	
Unknown/missing	1 (0.9)	0 (0.0)	1 (14.3)	
Clinical stage, n (%)				
I	7 (6.5)	1 (9.1) *	5 (71.4) *	0.000
II	4 (3.7)	1 (9.1)	0 (0.0)	
III	64 (59.8)	4 (36.4)	1 (14.3)	
IV	32 (29.9)	2 (18.2)	0 (0.0)	
Unknown/missing	0 (0.0)	3 (27.3)	1 (14.3)	
CCI, n (%)				
0	89 (83.2)	13 (92.9)	5 (71.4)	0.327
1	12 (11.2)	0 (0.0)	1 (14.3)	
2	3 (2.8)	1 (7.1)	1 (14.3)	
3	3 (2.8)	0 (0.0)	0 (0.0)	
Performance score, n (%)				
0	50 (46.7)	9 (75)	3 (42.9)	0.483
1	34 (31.8)	2 (16.7)	4 (57.1)	
2	19 (17.8)	1 (8.3)	0 (0.0)	
3	4 (3.7)	0 (0.0)	0 (0.0)	
CA125, U/mL	850 (352–1870)	425 (112–820) *	259 (122–560) *	0.007
RMI score	5463 (2196–12,150)	2460 (441–3860) *	840 (366–5040) *	0.004
Follow-up, months	38.4 (18.6–51.3)	39.7 (14.3–57.2)	67.6 (59.5–70.5) *	0.001
Mortality at endpoint, n (%)	73 (68.2)	7 (50.0) *	0 (0.0) *	0.000

Baseline and follow-up characteristics in women with ovarian cancer, other cancer, or borderline tumor. Other cancers included cervix cancer (*n* = 1), endometrial cancer (*n* = 5), teratoma (*n* = 1), neuroendocrine carcinoma (*n* = 2), and cancer of unknown origin (*n* = 5). Categorical variables are indicated as number (*n*) and percentage (%) of patients, and continuous variables are indicated as median (25th percentile; 75th percentile). *p*-value: comparison of all groups using one-way ANOVA or Kruskal–Wallis test. * *p* < 0.05 when compared to the ovarian cancer group using Student’s *t*-test or Wilcoxon rank-sum test. BMI, body mass index; CA125, cancer antigen 125; CCI, Charlson Comorbidity Index; RMI, risk of malignancy index.

**Table 2 ijms-25-02014-t002:** IGF protein levels the circulation and ascites of women with ovarian cancer.

Protein (ng/mL)	Circulation	Ascites	Ratio (Ascites/Circulation)	*p*
IGF-1	81 (62–101)	92 (71–120)	1.12 (0.83–1.48)	<0.01
IGF-2	422 (335–566)	313 (220–468)	0.71 (0.56–0.89)	<0.001
IGF bioactivity	1.12 (0.89–1.33)	2.62 (1.31–3.32)	2.72 (1.13–3.20)	<0.001
IGFBP-4	206 (187–237)	48 (27–58)	0.23 (0.16–0.29)	<0.001
CT-IGFBP-4	69 (61–87)	131 (77–157)	1.64 (1.36–1.97)	<0.001
NT-IGFBP-4	148 (103–180)	298 (209–346)	1.99 (1.17–2.60)	<0.001
PAPP-A	0.89 (0.75–1.12)	45 (34–62)	51 (33–75)	<0.001
PAPP-A2	0.44 (0.22–1.05)	1.77 (1.24–2.62)	3.93 (2.19–7.78)	<0.001
STC1	1.08 (0.63–1.94)	2.12 (1.23–3.92)	1.90 (1.23–3.02)	<0.001
STC2	37 (28–48)	68 (40–89)	1.68 (1.27–2.45)	<0.001

Variables are median (25th percentile; 75th percentile). For IGFBP-4, IGFBP-4 fragments, and IGF bioactivity, levels were only determined in a subgroup of 20 patients. *p*-values represent a comparison between ascites and circulating levels. CT, C-terminal; IGF, insulin-like growth factor; IGFBP, IGF binding protein; NT, N-terminal; PAPP-A, pregnancy-associated plasma protein-A; STC, stanniocalcin.

**Table 3 ijms-25-02014-t003:** Relationships between the concentrations of IGF proteins in ascites and the circulation of patients with ovarian cancer.

		IGF-1	IGF-2	IGF Bioactivity	IGFBP-4	CT-IGFBP-4	NT-IGFBP-4	PAPP-A	PAPP-A2	STC1	STC2
IGF-1	r	**0.297**		0.558		−0.493	−0.544		−0.376		0.210
	*p*	**0.002**		0.000		0.027	0.001		0.000		0.021
IGF-2	r		**0.627**					0.248			
	*p*		**0.000**					0.006			
IGF bioactivity	r	*0.437*		**0.554**							
	*p*	*0.008*		**0.001**							
IGFBP-4	r										
	*p*										
CT-IGFBP-4	r	*0.319*		*0.500*	*0.485*	**0.686**	0.745				
	*p*	*0.048*		*0.041*	*0.048*	**0.001**	0.001				
NT-IGFBP-4	r	*0.330*		*0.506*		*0.606*	**0.668**				
	*p*	*0.045*		*0.040*		*0.010*	**0.001**				
PAPP-A	r	*0.180*	*0.223*							0.289	0.310
	*p*	*0.048*	*0.013*							0.003	0.001
PAPP-A2	r								**0.526**		
	*p*								**0.000**		
STC1	r	*0.317*	*0.259*	*−0.471*	*0.485*					**0.588**	
	*p*	*0.001*	*0.005*	*0.045*	*0.048*					**0.000**	
STC2	r			*−0.528*	*0.535*			*0.291*			**0.490**
	*p*			*0.029*	*0.001*			*0.002*			**0.000**

Correlation analyses of IGF proteins in ascites (italic font) and the circulation (regular font) in women with ovarian cancer. Correlation for the same protein in ascites and the circulation is shown in bold. Data are expressed as unadjusted correlation coefficient (Pearson’s correlation) r-values (top) and *p*-values (bottom). Only statistically significant correlations (*p* < 0.05) are reported. For correlations with IGFBP-4, IGFBP-4 fragments, and IGF bioactivity, analyses were only based on measurements in 20 patients. CT, C-terminal; IGF, insulin-like growth factor; IGFBP, IGF binding protein; NT, N-terminal; PAPP-A, pregnancy-associated plasma protein-A; STC2, stanniocalcin-2.

**Table 4 ijms-25-02014-t004:** Log-rank analysis of all-cause mortality.

Log-Rank Test	Low Protein Level (*n* = 53)		High Protein Level (*n* = 54)		*p*
	Range (ng/mL)	Mortality, n (%)	Range (ng/mL)	Mortality, n (%)	
**IGF-1**					
Circulation	62 [34–80]	41 (77.4)	101 [81–202]	32 (59.3)	0.013
Ascites	71 [31–90]	28 (52.8)	120 [92–211]	45 (83.3)	<0.001
**IGF-2**					
Circulation	335 [150–421]	28 (52.8)	565 [422–999]	45 (83.3)	0.005
Ascites	220 [111–312]	29 (54.7)	467 [313–890]	44 (81.5)	<0.001
**PAPP-A**					
Circulation	0.75 [0.16–0.89]	37 (69.8)	1.12 [0.89–2.1]	36 (66.7)	0.986
Ascites	34 [12–44]	40 (75.5)	60 [45–101]	33 (61.1)	0.989
**PAPP-A2**					
Circulation	0.22 [0.04–0.43]	31 (58.5)	1.04 [0.44–4.91]	42 (77.8)	0.013
Ascites	1.24 [0.18–1.70]	34 (64.2)	2.61 [1.77–84.4]	39 (72.2)	0.331
**STC1**					
Circulation	0.63 [0.04–1.01]	36 (67.9)	1.91 [1.08–17.2]	37 (68.5)	0.215
Ascites	1.23 [0.08–2.09]	34 (64.2)	3.69 [2.12–17.3]	39 (72.2)	0.138
**STC2**					
Circulation	28 [16–37]	37 (69.8)	48 [37–81]	36 (66.7)	0.826
Ascites	40 [16–68]	41 (77.4)	89 [68–235]	32 (59.3)	0.055

Number (n) and percentage (%) of events in groups based on median protein level split. *p*-values: log-rank test. IGF, insulin-like growth factor; PAPP-A, pregnancy-associated plasma protein-A; STC, stanniocalcin.

## Data Availability

The data that support the findings of this study are partly available on request from the corresponding author. The data are not publicly available in its full form as the General Data Protection Regulation (GDPR) applies.

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
