# Peer review of "The IGF–PAPP-A–Stanniocalcin Axis in Serum and Ascites Associates with Prognosis in Patients with Ovarian Cancer"

_ijms, 2024, doi:10.3390/ijms25042014_

Round 1

Reviewer 1 Report

Comments and Suggestions for Authors

Author Response

We thank the reviewer for thorough comments and highly appreciate the time and expertise that have been invested in the review. We have revised the manuscript according to the comments, which are also addressed below. Alterations within the manuscript are highlighted.

Comment:

The study aims to elucidate the role of specific proteins in the prognosis of ovarian cancer. Overall, the authors’ interest is the identification of possible biomarkers, which is always an interesting topic. In that regard, the reviewer thinks that probably in the discussion a sentence or two on the properties of biomarkers would be an interesting add-on. Overall, the study was well designed, performed, and the discussion/conclusion in-line with the results reported.

TITLE

The authors refer that this axis has implications in patients’ prognosis. The reviewer thinks that, probably, the authors can discriminate in the title if it is a bad prognosis.

Response: In general, high levels of the proteins are associated with poor outcomes. However, for serum IGF-1, this is not the case since overall survival was longer for patients with high IGF-1. Thus, unfortunately, we cannot state in the title, that the axis is associated solely with a worse prognosis, as it is protein-dependent.

Comment:

ABSTRACT

The abstract is well written. The authors start by stating the importance of such proteins, and then disclose previous work. Then, they introduce the importance of the study are present the main results. Finally, they declare the importance of the findings and potential clinical implications. A reader will understand perfectly the rationale behind the study.

Line 20: the word [range] seems to be misplaced

Response: Thank you for this observation. We have moved [range] to a more suitable position.

Comment:

INTRODUCTION

The introduction is clear and with the necessary information to follow up the study developed.

Line 35: define the meaning of vague symptoms

Response: We have rephrased, now using ‘often limited and unspecific symptoms’.

Comment:

MATERIALS AND METHODS

This section is well detailed.

Study cohort and patient follow-up: The reviewer thinks that it might be interesting for the reader to have a diagram with the initial number of recruited patients and then the causes for exclusion, ending up with the total of samples analyzed.

Response: Thank you for this suggestion. We have included a figure illustrating the flow chart of data collection in the study (Fig. 4).

Comment:

RESULTS

The results are clear and well presented. Only minor comments are listed.

Line 79: This information might be useful in table 1.

Line 104: The reviewer does not know if this would be possible; but at least the circulating levels of these proteins in healthy people would be an interesting information to compare with patients.

Line 177: It would be possible or it make sense to present the mortality in percentage instead of N? The reviewer thinks that is easier to compare between groups.

Response:

Thank you for your suggestions for improvements.

Line 79: We have added information on treatment to Table 1.

Line 104: It would indeed be interesting to compare patient levels to those of healthy subjects. However, in this study, we did not have access to a representative control cohort, which should at least be sex- and age-matched. However, in our previous pilot study, we included 15 age-matched healthy women, and this information has now been added to the current manuscript limitations (l. 366).

Unfortunately, in the original study, we did not determine IGFBP-4, PAPP-A2, STC1, and STC2 in the controls, and thus, we cannot compare to the present results. Furthermore, the immunoassays for these proteins are less standardized and validated, so we would be cautious to compare our protein levels to those obtained in other studies using different setups, different days, different lab equipment etc.

Line 177: This is a great suggestion, and we have added percentages to the Table.

Comment:

DISCUSSION

The discussion is clear and holds on the results reported.

It would be possible to include one or two sentences on the definition of a biomarker. In addition, which characteristics should they have? Because it is always a sensitive topic, and some people do not know what makes a protein a good biomarker.

Response: This is a nice suggestion, and we agree with the reviewer; the definition and understanding of biomarkers are sometimes limited. We have elaborated in the discussion (l. 246) and have added a description of how the prognostic and predictive power of these biomarkers, as well as any gain in performance beyond that of conventional clinical risk factors, should be further validated in future independent validation cohorts (p. 353).

Comment:

CONCLUSION

The section is clear and supported by the results.

Response: Thank you.

Reviewer 2 Report

Comments and Suggestions for Authors

The manuscript by Rikke Hjortebjerg and co-authors describes the levels of different proteins (IGF-1, IGF-2, PAPP-A, PAPP-A2, STC1, STC2, and others) in serum and ascites of patients with ovarian cancer. A thorough statistical analysis was carried out concerning the interrelations of the protein levels, IGF bioactivity, and the patient survival. It was found that the IGF system presence and activity in ascites  are significantly different from that in circulation. Importantly, the authors showed that the investigated protein levels can be used for the prognosis of the patient outcomes. The manuscript deserves publication.

Specific comments:

Table 1:  A more informative table caption is needed.

Figure 3:  In the figure caption, it is necessary to explain the meaning of the dashed lines.

Summarizing, I recommend acceptance of the manuscript for publication after minor revision.

Author Response

We thank the reviewer for these nice comments, and especially for the time and expertise that have been invested in the review. We have revised the manuscript according to the comments, and alterations are highlighted in the manuscript.

Comments and Suggestions for Authors

The manuscript by Rikke Hjortebjerg and co-authors describes the levels of different proteins (IGF-1, IGF-2, PAPP-A, PAPP-A2, STC1, STC2, and others) in serum and ascites of patients with ovarian cancer. A thorough statistical analysis was carried out concerning the interrelations of the protein levels, IGF bioactivity, and the patient survival. It was found that the IGF system presence and activity in ascites are significantly different from that in circulation. Importantly, the authors showed that the investigated protein levels can be used for the prognosis of the patient outcomes. The manuscript deserves publication.

Specific comments:

Table 1:  A more informative table caption is needed.

Figure 3:  In the figure caption, it is necessary to explain the meaning of the dashed lines.

Summarizing, I recommend acceptance of the manuscript for publication after minor revision.

Response:

Thank you for these suggestions.

Table 1: We have tried to elaborate on the table caption. If the reviewer has any specific missing information in mind, please let us know.

Figure 3 caption: We have clarified the meaning of the dashed line.